The past century of coral bleaching in the Saudi Arabian central Red Sea

DeCarlo Thomas M. thomas.decarlo@kaust.edu.sa
Red Sea Research Center, Division of Biological and Environmental Science and Engineering, King Abdullah University of Science and Technology , Thuwal , Saudi Arabia
College of Natural and Computational Sciences, Hawaii Pacific University , Honolulu , HI , United States of America
Banaszak Anastazia
Electronic publication date: 2020 Oct 23
Publication date: 2020
Volume: 8
Electronic Location ID: e10200
Received 2020 Jul 14; Accepted 2020 Sep 26
Copyright: ©2020 DeCarlo
Copyright year: 2020
Copyright holder: DeCarlo
License: This is an open access article distributed under the terms of the Creative Commons Attribution License, which permits unrestricted use, distribution, reproduction and adaptation in any medium and for any purpose provided that it is properly attributed. For attribution, the original author(s), title, publication source (PeerJ) and either DOI or URL of the article must be cited.
License URL: https://creativecommons.org/licenses/by/4.0/

Keywords: Coral, Bleaching, Sea surface temperature, Coral cores, Ocean warming

Funding: Support was provided by the King Abdullah University of Science and Technology (KAUST) through the Red Sea Research Center and baseline funding to Michael L. Berumen. The funders had no role in study design, data collection and analysis, decision to publish, or preparation of the manuscript.

==============================
Accurate knowledge of the spatial and temporal patterns of coral bleaching is essential both for understanding how coral reef ecosystems are changing today and forecasting their future states. Yet, in many regions of the world, the history of bleaching is poorly known, especially prior to the late 20th century. Here, I use the information preserved within skeleton cores of long-lived Porites corals to reconstruct the past century of bleaching events in the Saudi Arabian central Red Sea. In these cores, skeletal “stress bands”—indicative of past bleaching—captured known bleaching events that occurred in 1998 and 2010, but also revealed evidence of previously unknown bleaching events in 1931, 1978, and 1982. However, these earlier events affected a significantly lesser proportion of corals than 1998 and 2010. Therefore, coral bleaching may have occurred in the central Red Sea earlier than previously recognized, but the frequency and severity of bleaching events since 1998 on nearshore reefs is unprecedented over the past century. Conversely, corals living on mid- to outer-shelf reefs have not been equally susceptible to bleaching as their nearshore counterparts, which was evident in that stress bands were five times more prevalent nearshore. Whether this pattern of susceptible nearshore reefs and resistant outer-shelf reefs continues in the future remains a key question in forecasting coral reef futures in this region.

Introduction

Reef-building corals thrive in the warm waters of the tropical oceans. Yet, despite their affinity for warm regions, anomalous increases in temperature of as little as 1 °C can damage the symbiosis between the coral host and the photosynthetic algae that inhabit the host’s tissues. When summertime temperatures exceed typical levels, the endosymbiont photosystem is impaired and begins to produce more reactive oxygen species than the coral can tolerate, leading to the expulsion of the symbionts (Lesser, 1997; Baker, Glynn & Riegl, 2008). Without their pigmented symbionts, corals turn white as their calcium carbonate skeletons become visible through their translucent tissues. Since the symbionts provide most of the energy that the coral needs to survive, bleaching causes reduced growth, fecundity, and lipid content, eventually leading to mortality if bleaching is prolonged (Mendes & Woodley, 2002; Grottoli, Rodrigues & Juarez, 2004). Coral bleaching was first observed by scientists in 1929 during the Great Barrier Reef Expedition on a shallow reef flat when calm weather led to localized heating of reef waters (Yonge & Nicholls, 1931). Other small-scale bleaching events were later observed due to localized disturbances such as freshwater (Goreau, 1964) or cold stress (Muscatine, Grossman & Doino, 1991; Saxby, Dennison & Hoegh-Guldberg, 2003; Hoegh-Guldberg & Fine, 2004; Paz-García, Balart & García-De-Léon, 2012), with heat-induced, ocean basin-scale mass coral bleaching first observed in 1982/1983 (Glynn, 1983; Glynn, 1993; Coffroth, Lasker & Oliver, 1990), although other events may have occurred earlier (Cavole & DeCarlo, 2020). Since 1982, global mass coral bleaching events have increased in frequency, most likely caused by more common heat stress due to global warming (Hughes et al., 2017; Hughes et al., 2018; Sully et al., 2019).

The Red Sea is lined by thousands of kilometers of near-continuous, shallow-water coral reefs (Rowlands et al., 2012; Churchill et al., 2019). These reefs are mostly restricted to the continental shelf, ranging from fringing reefs along the coast to knoll reef platforms dotting the shelf until the slope break (Montaggioni et al., 1986). Although the Red Sea is one of the warmest coral reef regions with summertime temperatures in the central and southern Red Sea routinely reaching 33 °C, corals living there are not necessarily more prone to bleaching than conspecifics in cooler regions because they have acclimatized or adapted to their local environment. Experimental evidence suggests that corals in the far northern Red Sea, however, are currently living as much as 4 °C below their thermal maximum, potentially a result of the south to north (i.e., warmer to cooler) migration of corals following the most recent glaciation (Fine, Gildor & Genin, 2013; Krueger et al., 2017). This has led to suggestions that the far northern Red Sea could be a climate change refuge for at least some species (Fine, Gildor & Genin, 2013; Kleinhaus et al., 2020), a hypothesis that—so far—has proven true since no bleaching has been observed there (Osman et al., 2018). Conversely, the central Red Sea has experienced several bleaching events, which were directly observed in 1998 (Devantier et al., 2000; Devantier & Pilcher, 2000), 2010 (Furby, Bouwmeester & Berumen, 2013; Pineda et al., 2013), and 2015 (Monroe et al., 2018). Yet, there were few systematic reef surveys prior to 1998 in the Red Sea (but see Antonius, 1988), leaving gaps in our knowledge of whether earlier bleaching events occurred in this region.

Although the surface waters of the Red Sea have warmed by ∼0.8 °C since 1900, in line with the global average for coral reef regions (Lough, Anderson & Hughes, 2018), the warming trend has been punctuated by sporadic episodes of anomalously high temperatures and pulses of rapid warming (Raitsos et al., 2011; Krokos et al., 2019). In particular, two relatively warm intervals, one centered around the 1930s and the other around the 1970s, are superimposed on the centennial warming trend (Krokos et al., 2019). These warm periods result from the combination of climate teleconnections in the north Atlantic and equatorial Pacific (Karnauskas & Jones, 2018; Krokos et al., 2019). Critically, since systematic surveys of coral health had not yet begun in the Red Sea, we do not know if these earlier warm periods were associated with coral bleaching events, or alternatively, if the recent bleaching events in the central Red Sea are unprecedented. Here, I use the stress history preserved within the skeletons of long-lived Porites spp. corals from the Saudi Arabian central Red Sea to reconstruct bleaching events over the past century, and I compare the presence of stress bands to a variety of ocean temperature and wind datasets.

Methods

Bleaching histories from coral skeletal cores

Between August 2019 and January 2020, I drilled twenty-two skeletal cores from massive Porites spp. coral colonies near the Thuwal region of the central Red Sea (Fig. 1). The primary study location was a nearshore reef called Abu Shosha (11 cores), but I also collected cores at reefs across the shelf (1-2 cores per reef, 11 total cores). The cores were drilled downwards from the top of each colony using a pneumatic drill with either 3- or 5-cm diameter diamond-tipped bits. All cores were rinsed several times with fresh water, then with pure ethanol, and air dried.

Figure 1 Study region in the Saudi Arabian central Red Sea.

(A) Red Sea coastlines (black) from the GSHHG database (Wessel & Smith, 1996), and 100-m isobath (Amante & Eakins, 2009) to delineate the shelf edge (gray). (B–C) Satellite imagery (ESRI/Maxar) of the Thuwal region (B) and Abu Shosha (C), with coral drilling locations shown as pink dots and the shelf edge (100-m isobath) indicated by the dashed white line.

To visualize density banding patterns within the cores, I scanned them with computerized tomography (CT) using a General Electric BrightSpeed machine. Scan settings included 100 kV voltage, 9.6 cm field of view, and 1.25 mm pixel spacing (i.e., resolution). The CT scans were visualized using Osirix software, which enabled me to cut digital slices 3–5 mm thick along the primary growth axes (Carilli et al., 2009; Cantin et al., 2010). Annual density bands and anomalous high-density “stress bands” were identified visually based on the banding pattern of each coral. Most of the cores had clear annual banding patterns with little ambiguity. Stress band identifications were based on their previously described appearances as sharply-defined bands that are anomalous relative to the regular annual banding pattern (Hudson et al., 1976; Carilli et al., 2009; Cantin & Lough, 2014; Barkley & Cohen, 2016; DeCarlo et al., 2017; DeCarlo et al., 2019; Barkley et al., 2018). The cores were not identified to species level, however there are only three possible morphospecies (lutea, lobata, and solida) with massive growth form in this region (Terraneo et al., 2019), and a previous study found no effect of species on stress band prevalence (DeCarlo et al., 2020).

I followed the statistical approach of Hendy, Lough & Gagan (2003) to compare the prevalence of stress bands among years. Specifically, I calculated the probability of finding, per year, the number of observed stress bands or fewer based on bleaching events affecting a certain number of Porites corals. Determining these probabilities enables statistical testing of severity between events (e.g., whether the number of observed stress bands in a given year is statistically different from another year, or from zero). These calculations require three inputs: the number of replicates (i.e., the number of cores covering each year), the number of stress bands observed per year, and the severity of hypothetical bleaching events. Since the latter is arbitrary, I repeated the calculations using two severity levels: the proportion of Porites that were bleached on Abu Shosha (∼1/3) during underwater surveys in 2015 (Monroe et al., 2018), and the maximum proportion of stress bands observed in any one year (see Results).

Climate data

I used four sea surface temperature (SST) datasets that differ in temporal coverage and spatial resolution. Two daily satellite-based products, Optimum Interpolation SST (OI-SSTv2) (Reynolds et al., 2007; Banzon et al., 2016) and Coral Reef Watch’s CoralTemp (CRW) (Liu et al., 2014) cover from present back until 1982 and 1985, at spatial resolutions of ∼25 km and ∼5 km, respectively. To assess temperatures prior to the satellite era, I used HadISST (Rayner et al., 2003) and ERSSTv5 (Huang et al., 2017), which are both monthly and have spatial resolutions of 1° (∼110 km) and 2° (∼220 km), respectively. I calculated degree heating weeks (DHW), a metric of heat stress that incorporates both the magnitude and duration of SST anomalies exceeding the maximum monthly mean (MMM) (Liu, Strong & Skirving, 2003), separately for each dataset. For the two monthly datasets (HadISST and ERSSTv5), I adjusted the typical DHW formula such that DHW begin to accumulate as soon as SST exceeds the MMM, rather than requiring SST to exceed the MMM by 1 °C. This was done because it is rare that the mean SST of an entire month is 1 °C greater than the MMM (occurring only once for each dataset between 1930 and 2019). Thus, the magnitudes of DHW calculated from HadISST and ERSSTv5 are not directly akin to those of OI-SSTv2 and CRW, although relative differences between years are comparable. For all datasets, I extracted the time series of the single grid-box closest to our primary study site, Abu Shosha. Finally, for each SST dataset, I calculated the maximum DHW that occurred per year for comparison to bleaching histories.

To assess the quantity of shipboard observations upon which HadISST and ERSSTv5 are based during the pre-satellite era, I downloaded SST measurement metadata from the International Comprehensive Ocean-Atmosphere Data Set (ICOADS3.0.0). I counted the number of observations in ICOADS corresponding to select bleaching years that fell within the HadISST and ERSSTv5 grid-boxes nearest to Abu Shosha.

Wind speed data were acquired from the European Centre for Medium-Range Weather Forecasts ERA5 reanalysis product (Copernicus Climate Change Service (C3S), 2017). The data were downloaded as monthly means at 31-km resolution covering 1979-2019, and the average August-September wind speed anomalies were calculated per year for the grid-box nearest to Abu Shosha. August and September were selected because these are the months when coral bleaching has been observed in this region (Devantier & Pilcher, 2000; Pineda et al., 2013; Monroe et al., 2018).

Photosynthetically active radiation (PAR) data were acquired from the moderate resolution imaging spectroradiometer (MODIS) onboard the Terra satellite (2000–2018) and from SeaWIFS (1998–1999) (Frouin, Franz & Werdell, 2002). Like the wind data, I averaged the August-September PAR anomalies per year for the grid-box nearest to Abu Shosha.

Results

The oldest observable bands in the coral skeletal cores from Abu Shosha ranged from 1917 to 1986 with a median age of 1966 (Fig. 2), and those in cores from outside of Abu Shosha ranged from 1907 to 2010 with a median age of 1968 (Fig. 3). In total, I counted 1,244 annual bands. At Abu Shosha, I identified high-density stress bands in 2015 (1/11 cores), 2010 (8/11 cores), 1998 (8/11 cores), 1982 (3/10 cores), 1978 (1/10 cores), and 1931 (1/2 cores) (Fig. 2). Only one core (K15) had no stress bands in any year. Most stress bands were distinct high-density anomalies, but their appearance varied among cores. For example, the 2010 stress bands in cores K24 and K2 were the most obvious because both the absolute density of the stress band, and the sharpness of the density gradient leading into the stress band, were exceptional relative to the rest of the core. Most other identified stress bands fit within these same criteria, but with weaker anomalies than K24 and K2. Conversely, some stress bands (e.g., those in core K1) were identified based on anomalies to the normal banding pattern, but without exceptionally high density. Both the 1998 and 2010 stress bands in core K1 are characterized by distinct, thin bands that disrupt the regular, broad annual bands. Although this type of stress band was rare in the dataset, similar stress bands have been linked to directly-observed bleaching in other studies (DeCarlo & Cohen, 2017). In a few cases, stress bands appeared to result in multi-year changes in density or banding patterns (e.g., K20 after 1998, and K13 after 1982). Outside of Abu Shosha, I found only three stress bands among the 11 cores (two of which were in the same core), corresponding to 1948, 1978, and 2010 (Fig. 3). Details of all coral coring locations, as well as age and stress bands results, and listed in Supplemental Table S1.

Figure 2 CT scans of coral skeletal cores from Abu Shosha.

Light (dark) shading indicates relatively high (low) density. The brightness and contrast of each core was adjusted separately to best visualize banding patterns, meaning that shading is comparable within, but not between, cores. Yellow lines connect across cores the first year (2019) and previous decades beginning with 2000. The oldest band of each core is also indicated. Dashed white lines connect stress bands across cores. Core ID numbers are shown at the top in white text. Inset boxes show higher magnification images of stress bands, grouped by year. In some cases, a different digital slice from the main core image was selected to better view the stress bands. Arrows and white lines between cores in inset images indicate the stress bands. The 10-cm scale bar in the main figure applies to all cores, whereas each image in the stress-band inset boxes has its own 1-cm scale bar.

Figure 3 CT scans of coral skeletal cores collected outside of Abu Shosha.

Light (dark) shading indicates relatively high (low) density. The brightness and contrast of each core was adjusted separately to best visualize banding patterns, meaning that shading is comparable within, but not between, cores. Yellow lines connect across cores the first year (2019) and previous decades. The oldest band of each core is also indicated. Core ID numbers are shown at the top in white text. Inset boxes show higher magnification images of stress bands. In some cases, a different digital slice from the main core image was selected to better view the stress bands. Arrows in inset images indicate the stress bands. The 10-cm scale bar in the main figure applies to all cores, whereas each image in the stress-band inset boxes has its own 1-cm scale bar.

I calculated the probability of finding the observed number of stress bands or fewer for each year of the two time-series (Abu Shosha, and outside of Abu Shosha). These calculations were performed for a bleaching event affecting either 1/3 (as observed in benthic surveys of Abu Shosha in 2015) or 8/11 (the number of stress bands identified in Abu Shosha corals during 1998 and 2010) of Porites colonies (Fig. 4). At Abu Shosha, I can exclude (with at least 95% confidence) that a comparable event to 1998/2010 occurred between 1934-2019 (Fig. 4A). Even though stress bands were also present in 1978, 1982, and 2015, the number of cores with stress bands in these years relative to the total number of cores was low enough that I could exclude an event affecting at least 8 out of 11 Porites. Conversely, even though only one stress band was associated with 1931, I cannot exclude that an event equivalent to 1998/2010 occurred in this year due to the small sample size. Detecting an event affecting 1/3 of Porites requires greater statistical power, and all years prior to 1970 cannot be excluded due to sample size. Additionally, based on stress band counts, I cannot exclude with 95% confidence that bleaching in 1978, 1982, or 2015 affected 1/3 or fewer of Porites colonies.

Figure 4 Probabilities of finding the observed number of stress bands or fewer per year.

The results are shown separately for Abu Shosha (A) and outside of Abu Shosha (B), for bleaching events affecting 1/3 (blue) or 8/11 (red) Porites corals. The dashed black lines indicate a probability of 0.05. Time periods covered by only a single core are shown in lighter shading. Bars lower (higher) than the 0.05 line indicate that we can (cannot) conclude with 95% confidence that there are fewer observed stress bands in that year than expected for the given proportion (1/3 or 8/11) of corals affected by a hypothetical event. (C) Simple proportion of corals with stress bands each year, displayed separately for Abu Shosha and outside of Abu Shosha, and with years covered by three or less total cores displayed in lighter shading.

Figure 5 Heat stress histories of the Saudi Arabian central Red Sea.

Maximum annual DHW are shown for the monthly HadISST (maroon) and ERSSTv5 (mustard) datasets (A), and the daily OI-SSTv2 (brown) and CRW (turquoise) datasets (B). Years with stress bands are labelled. Inset boxes show the correlations between maximum annual DHW calculated from the two pairs of datasets.

Outside of Abu Shosha, cores were from younger corals, on average, compared to Abu Shosha, which limits the statistical power (Fig. 4B). Nevertheless, it is clear that the 1998 and 2010 events that affected most Porites at Abu Shosha did not similarly affect other reefs, and I can exclude that there were any events affecting 8 out of 11 Porites since 1949. The 2010 and 1978 stress bands found in cores K9 and K6, respectively, suggest that these bleaching events did have at least some influence on corals outside of Abu Shosha, though. Conversely, the 1948 stress band in core K6 is not replicated in either the two other cores from outside Abu Shosha, or in any of the four Abu Shosha cores, that extend that far back in time.

The four SST datasets used here show some similarities in heat stress during overlapping years, but with key differences (Fig. 5). According to HadISST, the greatest DHW was reached in 1931 (6.0 °C-weeks), followed by 1998 (5.4 °C-weeks) (Fig. 5A). However, 1978 and 1982 had no heat stress, and DHW in 2010 and 2015 were lower than several other years without bleaching or stress bands. Conversely, ERSSTv5 shows only modest heat stress in 1931 (2.8 °C-weeks) and 1998 (3.6 °C-weeks), but much higher DHW in 2010 (9.5 °C-weeks) and 2015 (8.9 °C-weeks). Similar to HadISST, there is no heat stress in ERSSTv5 during 1978 or 1982, and there are several years with relatively high heat stress (>4 °C-weeks) but without any evidence of bleaching (e.g., 1941, 1957, 1963, 2001, 2006, 2017). Higher-resolution satellite SST datasets (OI-SSTv2 and CRW) likewise show some agreement, but also key disparities in certain years (Fig. 5B). For example, both OI-SSTv2 and CRW show no heat stress in 1982, nearly the same DHW in 1998 (4.6 and 4.4 °C-weeks, respectively), and relatively low DHW in 2015. However, the two depart during 2010, when OI-SSTv2 shows only 0.5 °C-weeks while CRW reached its highest DHW for this location of 5.3 °C-weeks. During 2010, CRW displays highest DHW nearshore (e.g., Abu Shosha), but with at least 3 ° C-weeks on the mid- to outer-shelf reefs (Fig. 6). Thus, even though the lower spatial resolution of OI-SSTv2 means that it would average some of these spatial differences of DHW (Fig. 7D), this alone cannot explain the full difference between the two datasets. Overall, the correlation (r2) between DHW calculated with the two longer-term datasets that are based primarily on shipboard measurements is only 0.16, and that between the two satellite-based datasets is even worse (0.13) (Fig. 4).

Figure 6 Spatial distribution of maximum DHW in 2010 according to CRW.

Black circles indicate Abu Shosha and the sampling location of each core collected outside of Abu Shosha.

Figure 7 Historical shipboard SST measurements in the ICOADS database.

(A–C) Locations and numbers of shipboard SST measurements in August and September of 1931, 1982, and 1998. Each measurement is plotted as a semi-transparent gray circle, such that darker gray coloring indicates multiple measurements in the same location. Numbers of measurements (n) are displayed for the central Red Sea (the area covered in the maps), and both the HadISST and ERSSTv5 grid-boxes nearest to Abu Shosha. (D) The size of grid-boxes in the OI-SSTv2 and CRW datasets. We use the OI-SSTv2 grid-box that covers Abu Shosha in our calculations, but the next grid-box to the west is also displayed to visualize its relation to shelf and offshore water, and both grid-boxes produce similar results such that the choice of grid-box does not affect our conclusions.

During August and September of 1931, there were 544 shipboard measurements in the central Red Sea, 274 of which were located within the ERSSTv5 grid-box, and only 1 of which was located within the HadISST grid-box (Fig. 7). By 1982, these numbers increased to 863, 535, and 26, respectively. Finally, in 1998, the sampling intensity was even greater, with 4,644 measurements in the central Red Sea, 2,466 in the ERSSTv5 grid-box, and 742 measurements in the HadISST grid-box.

Wind speeds during August and September of years with observed bleaching or stress bands on Abu Shosha were all anomalously low, although not exceptionally so (Fig. 8A). The lowest wind speeds between 1979 and 2018 occurred in 2003, followed by 1982. The years of highest PAR were 1998, 2002, and 2004 (Fig. 8B). Although the bleaching year of 2010—like 1998—was also associated with a positive PAR anomaly, PAR during 2015 was unusually low.

Figure 8 Wind and PAR anomalies near Abu Shosha.

(A) Mean wind speed anomalies for August–September of each year relative to the climatological mean August–September wind speeds of the full time series. Bleaching years are indicated with asterisks. (B) Same as (A) but for PAR anomalies.

Discussion

Scientists directly observed and reported coral bleaching in the central Red Sea during August–September of 1998, 2010, and 2015 (Devantier & Pilcher, 2000; Furby, Bouwmeester & Berumen, 2013; Pineda et al., 2013; Monroe et al., 2018). The current knowledge, which has been incorporated in a global assessment of spatial and temporal patterns of bleaching, is that no other bleaching events have occurred in this region, at least since 1980 (Hughes et al., 2018). It should be noted that coral disease surveys began near Jeddah (∼100 km to the south) in 1982 and continued for several years (Antonius, 1988). In these surveys, “frequent” coral bleaching was observed near Jeddah (13-25 colonies per 30-minute survey), although the year(s) in which this occurred is not stated (Antonius, 1988). Since few colonies were affected and the author attributed the bleaching near Jeddah to a combination of pollution and freshwater run-off (albeit with no direct evidence), subsequent studies have not considered these early observations to represent heat-induced coral bleaching (Furby, Bouwmeester & Berumen, 2013; Hughes et al., 2018; Berumen et al., 2019).

In this study, stress bands in long-lived Porites corals are broadly consistent with this current knowledge, except that I also found three stress bands corresponding to 1982 at a nearshore reef, Abu Shosha (Fig. 2). Thus, the cores show evidence of 1982 bleaching that may be concurrent with direct observations of bleaching near Jeddah, even though the cores were collected ∼100 km from the pollution and freshwater sources. The 1982 event at Abu Shosha was significantly less severe than 1998/2010 (95% confidence), but I cannot exclude that minor bleaching of one third or fewer Porites corals occurred (Fig. 4). Additionally, I found two stress bands in 1978, and single stress bands in each of 1948 and 1931. The 1978 event is also significantly less severe than 1998/2010, but the 1931 and 1948 events cannot be differentiated from 1998/2010 without more samples. However, while 1978 had no heat stress, 1931 was the summer with highest DHW according to HadISST (Fig. 5). It is important to recognize that both the temperature and bleaching severity in 1931 are poorly constrained due to small sample sizes as I collected only three cores this old and there were far fewer shipboard temperature measurements in the 1930s than recent decades (Deser et al., 2010) (Fig. 7). Nevertheless, even though each is sample-size limited, the correspondence between the clear 1931 stress band in core K13 and the exceptionally high heat stress during summer 1931 in HadISST leaves open the possibility that this was a bleaching event.

Porites are generally considered to be relatively resistant to stress (Darling et al., 2012; McClanahan et al., 2020) and have shown evidence of acclimatization to repeated marine heatwaves (DeCarlo et al., 2019), meaning that they are not necessarily representative of entire coral communities. It is also possible that some stress bands could arise from disturbances other than high temperatures, such as disease. Finally, Porites colonies could conceivably miss a bleaching event if calcification completely ceases. Nevertheless, Porites stress bands have captured known bleaching events across the Indo-Pacific (Cantin & Lough, 2014; Barkley & Cohen, 2016; DeCarlo et al., 2017; DeCarlo et al., 2019; DeCarlo et al., 2020; Barkley et al., 2018), and the presence of Porites stress bands has correlated with community-level bleaching responses (Barkley & Cohen, 2016; Mollica et al., 2019; DeCarlo et al., 2020). Thus, that the cores from Abu Shosha generally recorded the known bleaching events in 1998, 2010, and 2015 supports the accuracy of Porites stress bands for representing past bleaching events in this region.

Heat stress in the central Red Sea is highly variable among SST datasets. Further, no single SST dataset’s heat stress history accurately captures both the presence and absence of bleaching events. HadISST shows highest DHW in 1931 and 1998, consistent with the bleaching history, but 1978, 1982, 2010, and 2015 all have lower DHW than multiple non-bleaching years. ERSSTv5, in contrast, shows highest DHW in 2010 and 2015, but fails to separate all other bleaching years. OI-SSTv2 only shows relatively high DHW in 1998, but not 1982, 2010, or 2015. Finally, CRW has its highest DHW in 1998 and 2010, but does not separate 1982 or 2015 from non-bleaching years.

It is important to recognize both the different spatial scales of the SST datasets and the potential for temporal biases due to changing measurement intensities. HadISST grid-boxes are 1° x1°, which is close to half the width of the Red Sea, and the grid-box closest to Abu Shosha is outside of the main shipping route where most pre-satellite measurements were made (Fig. 7). Although ERSSTv2 grid-boxes are even larger, encompassing effectively the entire width of the Red Sea, the closest grid-box to Abu Shosha captures the main shipping route. The satellite products also differ in spatial resolution. OI-SSTv2 grid-boxes (0.25°) near Abu Shosha either include a large portion of land or a combination of shelf and offshore waters, whereas CRW (5 km) grid-boxes are small enough to include only shelf areas (Fig. 7D). Temporally, the measurement intensity of surface water temperatures from ships increased dramatically over time (Figs. 7A–7C), leaving greater uncertainty in earlier SST data. The satellite-based data suffer from a similar problem, as new satellites have come into operation over time. OI-SSTv2 attempts to minimize temporal biases by using only one type of sensor (the advanced very high resolution radiometer, AVHRR) at the expense of spatial resolution, whereas CRW uses more satellites to achieve greater spatial resolution at the expense of biases in some earlier years (Liu et al., 2014; Banzon et al., 2016; DeCarlo & Harrison, 2019; DeCarlo, 2020). Thus, both have certain advantages, with OI-SSTv2 being more stable over time but CRW isolating shelf waters, making it difficult to conclude which is most accurate at a given time without validation against independent datasets.

Discrepancies between DHW calculated from satellite-based SST datasets have also been reported for the Great Barrier Reef (GBR) of Australia (DeCarlo & Harrison, 2019), albeit to a lesser degree than observed here in the central Red Sea. However, unlike the GBR, temperature logger data from Red Sea coral reefs are, to my knowledge, unavailable prior to 2009 (Davis et al., 2011; Blythe, Da Silva & Pineda, 2011; Pineda et al., 2013; DeCarlo et al., 2016), and intermittent since then. Testing the accuracy of different SST datasets during only the past decade is unlikely to resolve this problem because the differences are not stationary in time, meaning that accuracy in recent years does not necessarily translate to accuracy in previous decades (DeCarlo & Harrison, 2019). In addition to discrepancies between satellite SST datasets over broad spatial scales (kilometers), relatively large differences in temperature means and variances (of up to several ∘C) can exist over just hundreds of meters within coral reef environments (Davis et al., 2011; Pineda et al., 2013; Palumbi et al., 2014; DeCarlo et al., 2017). The Abu Shosha cores were collected from the reef slope on the exposed side of the reef crest (Fig. 1), where diurnal cycles of heating and cooling are less than on top of the shallow reef flat (Davis et al., 2011), but localized heating can still differentiate the water temperatures to which these corals were exposed from the larger-scale SST represented by satellite products (Pineda et al., 2013). Together, these issues present a fundamental limitation to understanding the drivers of coral bleaching in the central Red Sea, especially on shallow reef flats, because there is a low degree of confidence in the magnitudes of heat stress that have sparked bleaching. Corals themselves can shed light on past SST variability, though, since the trace-element geochemistry of their skeletons is sensitive to temperature (Corrège, 2006; Nurhati, Cobb & Di Lorenzo, 2011). Previous analyses of temperature proxies in coral skeletal cores from Abu Shosha and Shi’b Nazar showed some of the highest summer temperature peaks in the early 2000s (D’Olivo et al., 2019; see also Murty et al., 2018), broadly consistent with the satellite data, especially OI-SSTv2.

In addition to temperature, other factors such as light and nutrients can modulate the bleaching susceptibility of corals (Brown, 1997; Dunne & Brown, 2001; Wooldridge, 2009; Cunning & Baker, 2012; Wiedenmann et al., 2013; Vega Thurber et al., 2014; Skirving et al., 2017; DeCarlo & Harrison, 2019; DeCarlo et al., 2020). I tested if wind speed or PAR anomalies correspond to bleaching events. Low-wind conditions can amplify heating on shallow reefs (Davis et al., 2011; DeCarlo et al., 2017), and unusually high PAR can exacerbate heat stress (Dunne & Brown, 2001; Skirving et al., 2017). Although bleaching in this study tended to occur in low-wind years, wind speed is not a unique predictor of bleaching since there are several years with weaker winds than bleaching years (Fig. 8A). Additionally, low-wind conditions are likely responsible for some of the positive SST anomalies, making it difficult to treat these two factors as independent. Likewise, PAR anomalies do not seem to be a primary driver of bleaching here since there was only a modest positive anomaly in 2010 and a negative anomaly in 2015 (Fig. 8B). Yet, it is possible that the exceptionally high PAR in 1998 contributed, at least in part, to the severe bleaching response in that year. There are no continuous nutrient datasets available in this area, but a study of summertime upwelling ∼300 km south in the Farasan Banks may offer some indication of upwelled nutrient supply (DeCarlo et al., 2020). Upwelling of high-nutrient Gulf of Aden Intermediate Water (GAIW) in the Farasan Banks is one of the main sources of nutrients to Red Sea surface waters, and traces of GAIW (10–20%) have been identified at the shelf edge in the central Red Sea (Churchill et al., 2014). Thus, it is conceivable that corals on Abu Shosha are occasionally exposed to elevated nutrients as traces of GAIW are mixed onto the shelf. Notably, June-August of 1982 and 2015 were among the strongest upwelling pulses in the Farasan Banks, both of which are bleaching years on Abu Shosha despite relatively low heat stress.

The past century of bleaching

Below, I summarize the bleaching history of the Saudi Arabian central Red Sea as inferred from coral cores and direct observations, in addition to the heat stress history during these bleaching years.

1931: SST may have been anomalously high, especially in August, but HadISST and ERSSTv5 disagree on the magnitude of the heat stress (6.0 and 2.8 °C-weeks, respectively). A single stress band was found corresponding to 1931, but only two cores from Abu Shosha extend this far back. Thus, 1931 is a potential bleaching year, but limitations in both the SST and stress band data prevent definitive conclusions.

1948: A single stress band was found in an outer-shelf core, but no other stress bands were found either in two other cores from mid- or outer-shelf reefs or in four cores from Abu Shosha. Neither HadISST nor ERSSTv5 show any heat stress at this time. Thus, 1948 is unlikely to have been a severe bleaching event, but rather was more likely to have been either a minor bleaching event or a non-temperature disturbance to this specific coral.

1978: Two stress bands were found in this year, one from an outer-shelf reef and another at Abu Shosha. The sample size of cores was sufficient to exclude that an event affecting 8 out of 11 Porites occurred from 1934 through 1978 at Abu Shosha, and from 1949 through 1978 outside of Abu Shosha. However, an event affecting 1 out of 3 Porites in 1978 could not be excluded, leaving open the possibility that a minor bleaching event affected reefs across the shelf. The 1978 stress bands in cores K6 and K14 are both clear, but there are no stress bands in the other 13 cores of this age (9 from Abu Shosha and 4 from other reefs). Both HadISST and ERSSTv5 show no heat stress as this time. Although the co-occurrence of two stress bands in this year lends support to this being a real bleaching event, it probably affected a relatively small proportion of corals, and the environmental driver of any bleaching in 1978 is unclear.

1982: This year marks the first occurrence of stress bands in multiple cores from one reef, similar to the results of a similar study on the northern GBR and in the Coral Sea (DeCarlo et al., 2019). Likewise, 1982/1983 is the first known large-scale bleaching event that stretched across the Pacific Ocean (Glynn, 1983; Oliver, 1985). Some bleaching was observed near Jeddah around this time, but key details such as specific times, places, and bleaching extent are lacking (Antonius, 1988). SST datasets, including HadISST, ERSSTv5, and the satellite-based OI-SSTv2 all show no heat stress during this year. However, satellite-based SST also showed very little heat stress on the GBR in 1982 (DeCarlo & Harrison, 2019) even though the majority of corals bleached, and more than half died, on most nearshore reefs that were monitored at the time (Oliver, 1985). Furthermore, coral cores in the southern Red Sea show 1982 stress bands, albeit in <15% of corals, despite similarly low heat stress, likely a result of nutrient-injection from the strong upwelling that preceded bleaching there (DeCarlo et al., 2020). Therefore, I suggest that 1982 was probably a minor bleaching event on Abu Shosha even though heat stress was apparently either low or absent. This could be a result of either inaccuracies in the SST datasets and/or the exacerbation of minor heat stress by upwelled nutrients early in the summer.

1998: This is the only year that all four SST datasets agree that there was relatively high heat stress. Additionally, 1998 is the first time that mass coral bleaching in the central Red Sea was observed and recorded by scientists (Devantier & Pilcher, 2000). Although it is unclear if Abu Shosha itself was surveyed at this time, up to 65% of corals bleached on nearshore reefs around Rabigh, only 50 km to the north (Devantier & Pilcher, 2000). Consistent with the visual observations, coral cores from Abu Shosha reveal evidence of severe bleaching, with clear stress bands in 8 out of 11 cores. I can conclude with 95% confidence that no prior event reached this severity since 1934. Skeletal geochemistry of Abu Shosha Porites was also perturbed during this event, indicative of disruptions to the normal calcification process (D’Olivo et al., 2019). The complete absence of stress bands in cores outside of Abu Shosha suggests this event may have been restricted to nearshore environments.

2010: Similar to 1998, coral bleaching was observed directly on nearshore reefs in the central Red Sea in 2010, and this is corroborated by stress bands in 8 out of 11 cores at Abu Shosha. However, unlike 1998, not all SST datasets agree that there was substantial heat stress, with unprecedented DHW in ERSSTv5 and CRW, but relatively low DHW in HadISST and OI-SSTv2. Previous investigations of this bleaching event suggest that it arose from an eddy that constrained warm water near the coast (Pineda et al., 2013), which is only detectable in CRW. Thus, this event likely rivalled 1998 in severity on Abu Shosha, but probably reached a lesser extent of the Red Sea than previous bleaching events in 1982 and 1998 (Furby, Bouwmeester & Berumen, 2013). Only one core from outside Abu Shosha contained a 2010 stress band, indicating that—like 1998—this event mostly affected nearshore reefs, but there may have been minor effects on other reefs across the shelf.

2015: Coral bleaching was again directly observed in 2015, including approximately one third of Porites colonies at Abu Shosha and Qita Al-Kirsh (Monroe et al., 2018), where cores K11 and K12 were collected. However, the coral cores barely registered this event, with only 1 out of 11 Abu Shosha cores containing a faint stress band. Although this result is not significantly different from expected if only 1/3 of Porites bleached, it represents a marked difference in the Porites response compared to 1998 and 2010. This suggests that either bleaching was more severe on Abu Shosha in 1998 and 2010 than 2015, or that these colonies acclimatized following their previous exposure to heat stress, similar to results from the GBR (DeCarlo et al., 2019). No cores outside of Abu Shosha contained 2015 stress bands, although the outer-shelf reefs had <10% bleaching at this time in benthic surveys, including Shi’b Nazar (cores K4 and K5) (Monroe et al., 2018). As discussed previously (Monroe et al., 2018), there was little heat stress in 2015, with only the low-resolution ERSSTv5 showing substantial DHW. Like 1982, it is possible that minor heat stress was exacerbated by upwelled nutrients since 2015 was one of the strongest GAIW upwelling years of the satellite era (DeCarlo et al., 2020).

Conclusion

Regardless of uncertainties in the precise DHW during each individual year, it is clear that heat stress events have become more common and of higher magnitude since 1998. Likewise, the three bleaching events during this time (1998, 2010, and 2015) left signatures in the nearshore coral skeletal cores that, when viewed as a whole, are unprecedented in terms of frequency and severity since at least the mid-1930s. One potential glimmer of hope is that the results here suggest these corals may have acclimatized after 1998 and 2010, such that they were less affected by bleaching in 2015 (i.e., they may have bleached but not formed stress bands), consistent with signs of acclimatization in other regions (Vargas-Ángel et al., 2001; Guest et al., 2012; Pratchett et al., 2013; Putnam & Gates, 2015; Gintert et al., 2018; Coles et al., 2018; DeCarlo et al., 2019). Additionally, mid- and outer-shelf reefs have been less sensitive to heat stress than those nearshore, with evidence of only minor bleaching events in the skeletal cores and no apparent increase in frequency or severity in recent decades. A similar pattern of highly susceptible nearshore corals and more resilient offshore corals has been reported in the Mesoamerican reef system (Baumann et al., 2019). As sea surface temperatures increase in the coming decades, the central Red Sea will be subjected to more frequent and intense heat stress events (Cantin et al., 2010; Van Hooidonk, Maynard & Planes, 2013; Hoegh-Guldberg et al., 2014). Corals living on nearshore reefs of this region are therefore likely to experience more severe bleaching events with diminishing return times, unless they can acclimatize or adapt fast enough. On the other hand, corals on outer-shelf reefs of the Saudi Arabian central Red Sea—like their counterparts in the northern Red Sea—have so far avoided severe bleaching, but we do not yet know whether this resistance will be maintained as the ocean continues to warm.

Supplemental Information

Supplemental Information 1 Coral core locations, depths, oldest counted bands, and any stress bands observed

All cores were collected from colonies of the genus Porites. Depth indicates the water depth of the top of each colony. The oldest band indicates the age of the bottom-most annual band of each core.

Click here for additional data file.

I thank Vincent Saderne, Anna Knochel, Alex Kattan, Walter Rich, Alyssa Bell, Claire Shellem, Aislinn Dunne, Ashlie McIvor, Irene Salines-Akhmadeeva, and Michelle Havlik for assistance in the field, and Michael Berumen for logistical support.

Additional Information and Declarations

Competing Interests

Author Contributions

Data Availability

The author declares there are no competing interests.

Thomas M. DeCarlo conceived and designed the experiments, performed the experiments, analyzed the data, prepared figures and/or tables, authored or reviewed drafts of the paper, and approved the final draft.

The following information was supplied regarding data availability:

Raw data and code are available at Code Ocean: DOI https://doi.org/10.24433/CO.8604403.v3.

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
