# Peer review of "The past century of coral bleaching in the Saudi Arabian central Red Sea"

_PeerJ, doi:10.7717/peerj.10200_

## Round 0.1 · original submission · Minor Revisions

Two expert reviewers have evaluated your manuscript and their comments can be seen below. As you can see both have made positive comments about the manuscript. However, both have suggestions to improve the manuscript.

Reviewer 1 ·

Basic reporting

The manuscript uses stress bands from massive Porites cores in the central Red Sea, along with climatology data, to identify past bleaching events in the area of the past 100+ years. The study complements more recent in situ observations of coral bleaching from the past ~ 20 years, and despite identifying possible undocumented bleaching events prior to the 1980s, concludes that the frequency and severity of bleaching events has increased in recent years.

The manuscript is well-written and I feel is suitable for publication providing some minor comments (highlighted below under ‘general comments’) are addressed – these include some comments that are merely suggestions to be addressed at the author’s discretion.

Overall, I think the manuscript is straightforward and will be a solid contribution to the field – I commend the author for their work.

Experimental design

The overall rational and design of the study are clear and sufficiently detailed.

While the number of cores is limited, I recognise the difficulty and environmental impact of collecting a large number of cores. Additionally, the author seems to have used an appropriate statistical approach to account for this and, generally, interprets the results with this limitation in consideration – though there are a one or two sections, detailed below, where I think a more conservative interpretation may be more appropriate.

Validity of the findings

While I am not familiar with the technical aspects of using stress bands on coral cores to indentify past bleaching events, the author has previously published peer-reviewed work using this approach and cites past studies in support of the methodology. I therefore see little reason to doubt the validity of the approach unless another expert reviewer expresses concern.

Additional comments

Line 15: Do you mean 20th century?

Lines 31-32: It is my understanding that the term ‘zooxanthellae’ is colloquial and no longer appropriate. I would suggest removing it on line 31 and changing it to ‘endosymbiont photosystem’, or something similar, on the following line.

Line 61: ‘a’ hypothesis

Line 68: There is a paper by Raitsos et al. (2011) in Geophysical Research Letters that reports a considerable increase in temperature in the region since the mid-1990s – it may be relevant to refer to it here or elsewhere in the paper.

Line 75: not sure the ‘currently’ is necessary.

Line 77-79: I think you should state here that you also analyse and compare long-term climatology datasets that include temperature and wind data - you don’t only use the coral cores to reconstruct the bleaching events.

Line 84 and throughout: While I understand, and agree with, the benefit of using the active voice, I do think the use of ‘my’ often is not appropriate and could be replaced with ‘the’. Notably, on line 252, you say ‘my bleaching history’ when referring to the bleaching history reconstructed in the study – which I think is incorrect.

Line 86: I would suggest stating the total number of cores collected outside of Abu Shosha.

Line 146: When you say ‘ranged in age’ do you mean that they dated as far back as the years stated?

Line 148-150: I think it would be helpful to say how many of the 11 cores did or didn’t have bands here? Just to get an idea of the overall proportion of cores/colonies that did bleach.

Same for the non-Abu Shosha cores below (would be good to state many cores were taken in total and how many experienced bleaching).

Line 162: How many cores were these three bands found on?

Line 205-206: Did this analysis yield a p-value regarding the significance (or non-significance) of these correlations?

Line 249: I think this should toned down further considering the meagre evidence/data available - it leaves open the possibility that a bleaching event occurred in this year. There really isn’t any evidence to support the occurrence of a severe bleaching event.

Line 300: You allude to it somewhat here, but I think a clear statement is missing in the discussion that indicates that the recorded stress bands may have been caused by something other than bleaching (even maybe disease or something else?), which may explain why you found stress bands on one or two colonies in years despite no evidence of high temperature/light or low wind in those years.

Line 325-327: This really is a personal/stylistic thing, but I don’t personally agree with the approach to this section. I think the author could do without this statement and simply discuss the timeline of bleaching over the last century chronologically without separating it by year.

E.g. ‘In 1931, SSTs appear to have…’

This is very much up to the discretion of the author, however.

Line 328: Shouldn’t it be ‘SSTs appear’ rather than ‘SST appears’? There was more than one recording of SST this year, no?

Figures: The figure legend for figure 3 could be written more formally – I would suggest something like ‘CT scans of coral skeletal cores collected outside of Abu Shosha. Refer to figure 2 legend for a full description of the figure.’

·

Basic reporting

This is a well-written manuscript based on 22 Porites cores drilled out of long lived corals in the central Red Sea. This is a fortune when considering the amount of data recorded in these cores. I was excited and very curious reading the title of the ms and the abstract because the potential is huge. I was a bit disappointed when discovering that the bleaching record of the past century is not so straightforward and that the use of these cores was limited to tracking stress bands rather than supporting it with a stable isotopic work and densitometry. As is, it leaves many open questions and of course, the cores can be used for further analyses.

Yet, the ms is informative and important for anyone working in this region.

The title is a bit misleading because in fact we learn that beyond recent, observed bleaching events (1998, 2010, 2015) stress bands may not be the best proxy or, maybe there was only a minor, local bleaching event with only a few cores recording stress. so little do we learn about the bleaching events in the last century.
proposed title: Can stress bands reliably report bleaching events in the Red Sea?
Anyway, I enjoyed reading it.

Experimental design

No issues with the sampling procedure and analyses.

L148- the author state the number of annual bands counted. Impressive but why is is this important or useful?
The Porites species issue: how many/which species were likely included in the study and what is known on the species sensitivity to thermal stress?
Is it possible to present the densitometric profile along the core and linear extension? I find it very interesting and I suppose the data for this exists?
I'm not entirely sure I understood the need in calculating the probability of finding stress bands. maybe elaborate further on this.
Good job comparing the different sources of temperature records for the region.

Validity of the findings

Findings are straightforward. Well presented and analyzed.

Additional comments

As mentioned above, I like the study but would have liked to see more analyses to decipher the source of stress (temp, nutrients, light and interactions between these parameters).
Consider looking at cores (images in papers) from other regions in the Red Sea, can you detect some annual bands that might be stress bands? in some cases (Klein et al. 1997) these have also the isotopic signature to support thermal stress.
Porties is considered thermally resilient and as such may be insensitive to high DHW. I think it is worth discussing it.
is it possible that under severe stressful conditions, the more sensitive porites colony will miss a band? not calcify/grow at all? if so, maybe some "missing stress bands" are the result of 0 growth?

---

## Round 0.2 · accepted · Accept

I am satisfied with the modifications made to the manuscript.